# Management of locally advanced non-small cell lung cancer in the modern era: A national Italian survey on diagnosis, treatment and multidisciplinary approach

**Alessio Bruni**[1]ᐦ*, **Niccolò Giaj-Levra**[2]ᐦ, **Patrizia Ciammella**[3], **Virginia Maragna**[4], **Katia Ferrari**[5], **Viola Bonti**[5], **Francesco Grossi**[6], **Stefania Greco**[7], **Carlo Greco**[8], **Paolo Borghetti**[9], **Davide Franceschini**[10], **Enrica Capelletto**[11], **Marco Perna**[4], **Giuseppe Banna**[12], **Stefano Vagge**[13], **Editta Baldini**[14], **Emilio Bria**[15], **Andrea Botti**[16], **Marcello Tiseo**[17], **Massimiliano Paci**[18], **Maria Taraborrelli**[19], **Venerino Poletti**[20,21], **Pierluigi Granone**[22], **Umberto Ricardi**[23], **Silvia Novello**[11], **Vieri Scotti**[4]

1 Radiotherapy Unit, Department of Oncology and Hematology, University Hospital of Modena, Modena, Italy, 2 Department of Advanced Radiation Oncology, IRCCS Sacro Cuore–Don Calabria Hospital, Negrar, Verona, Italy, 3 Radiation Therapy Unit, Department of Oncology and Advanced Technology, AUSL-IRCCS, Reggio Emilia, Italy, 4 Radiation Therapy Unit, Department of Oncology, Careggi University Hospital, Firenze, Italy, 5 Section of Respiratory Medicine, Careggi University Hospital, Firenze, Italy, 6 UOC Oncologia Medica Fondazione IRCCS Ca' Granda Ospedale Maggiore Policlinico, Milan, Italy, 7 UOSD of Oncologic Pneumology, San Camillo Forlanini Hospital, Rome, Italy, 8 Department of Radiation Oncology, Campus Bio-Medico University, Rome, Italy, 9 Radiation Oncology Department University and Spedali Civili, Brescia, Italy, 10 Department of Radiotherapy and Radiosurgery, Humanitas Cancer Center and Research Hospital, Rozzano, Milan, Italy, 11 Oncology Department, University of Turin, AOU San Luigi, Orbassano (TO), Italy, 12 Oncology Department, Ospedale Cannizzaro, Catania, Italy, 13 Department of Radiation Oncology, Azienda Ospedaliera Universitaria San Martino di Genova—IST, Istituto Nazionale Ricerca sul Cancro, Genoa, Italy, 14 UOC Oncologia Medica Ospedale San Luca, Lucca, Italy, 15 U.O.C. Oncologia Medica, Comprehensive Cancer Center—Fondazione Policlinico Universitario A. Gemelli, IRCCS—Università Cattolica del Sacro Cuore, Roma, Italy, 16 Medical Physics Unit, Department of Oncology and Advanced Technology, AUSL-IRCCS, Reggio Emilia, Italy, 17 Department of Medicine and Surgery, University of Parma and Medical Oncology Unit, University Hospital of Parma, Parma, Italy, 18 Division of Thoracic Surgery, Azienda USL-IRCCS of Reggio Emilia, Italy, 19 Radiotherapy Unit, SS Annunziata Hospital—G. D'annunzio University, Chieti, Italy, 20 Department of Diseases of the Thorax, Ospedale GB Morgagni, Forlì (I), Italy, 21 Department of Respiratory Diseases & Allergy, Aarhus University Hospital, Aarhus, Denmark, 22 Department of General Thoracic Surgery, Catholic University, Rome, Italy, 23 Department of Oncology, University of Turin, Torino, Italy

ᐦ These authors contributed equally to this work.

* brunialessio@virgilio.it

## Abstract

Concurrent chemotherapy and radiotherapy (cCRT) is considered the standard treatment of locally advanced non-small cell lung cancer (LA-NSCLC). Unfortunately, management is still heterogeneous across different specialists. A multidisciplinary approach is needed in this setting due to recent, promising results obtained by consolidative immunotherapy. The aim of this survey is to assess current LA-NSCLC management in Italy. From January to April 2018, a 15-question survey focusing on diagnostic/therapeutic LA-NSCLC management was sent to 1,478 e-mail addresses that belonged to pneumologists, thoracic surgeons, and radiation and medical oncologists. 421 answers were analyzed: 176 radiation oncologists, 86 medical oncologists, 92 pneumologists, 64 thoracic surgeons and 3 other

**Data Availability Statement:** All relevant data are within the paper and its Supporting Information files.

**Funding:** A pharma company (Astra Zeneca Italy, principal site in Basiglio, Milano, Italy) gave the unconditional support to develop a web platform for data entry. The funder had no role in study design, data collection and analysis, decision to publish, or preparation of the manuscript.

**Competing interests:** A pharma company (Astra Zeneca Italy, principal site in Basiglio, Milano, Italy) gave the unconditional support to develop a web platform for data entry. The funders had no role in study design, data collection and analysis, decision to publish, or preparation of the manuscript. No patents or products needs to be declared related to the study. This does not alter our adherence to PLOS ONE policies on sharing data and materials.

specialists. More than a half of the respondents had been practicing for >10 years after completing residency training. Some discrepancies were observed in clinical LA-NSCLC management: the lack of a regularly planned multidisciplinary tumor board, the use of upfront surgery in multistation stage IIIA, and territorial diffusion of cCRT in unresectable LA-NSCLC. Our analysis demonstrated good compliance with international guidelines in the diagnostic workup of LA-NSCLC. We observed a relationship between high clinical experience and good clinical practice. A multidisciplinary approach is mandatory for managing LA-NSCLC.

## Introduction

Concomitant radiotherapy and chemotherapy (cCRT) represents the standard of care in "fit patients" (defined as patients with good performance status, no or mild comorbidities, and potentially able to undergo a multimodal approach as reported in the RTOG0617 trial) with a diagnosis of locally advanced non-small cell lung cancer (LA-NSCLC) [1–2]. This combination is widely adopted due to strong, evidence-based results on overall survival (OS), progression free survival (PFS), and local control (LC) [3–4] when compared to a sequential approach. Despite the introduction of and improvements in image-guided radiotherapy (IGRT) (e.g. using pre-planned positron emission tomography–PET-CT) and radiation technologies (e.g. intensity modulated radiotherapy–IMRT), OS is still disappointing and generally unimpacted by the addition of surgery [5]. In particular, cCRT is characterized by a higher risk of developing acute and late toxicities (i.e. esophagitis and pneumonitis), and only 40% of LA-NSCLC patients are candidates for this approach [6]. Particularly, "fit patients" (e.g. perfomance status 0–1, median age < 65 years, Caucasian, few comorbidities) are reported to have interesting OS, such as in the RTOG0617 trial in which the control group had a median OS of 28.7 months and a 2-year OS of 57.6% [5]. Recently, good results were obtained using immunotherapy (IT) after cCRT with a significant improvement in PFS and OS, with no impact in severe acute/late toxicity profiles [5,7–8].

The management of LA-NSCLC seems to be inhomogeneous between oncological centers and specialists due to the challenges of selecting fit patients, the recent introduction of IT, and the consequentially differing expertise in LA-NSCLC management. The aim of this survey was to assess the management of LA-NSCLC in Italy, asking all involved specialists (pneumologists [PN], thoracic surgeons [TS], radiation [RO] and medical oncologists [MO]) so as to illuminate potentially critical issues, improve good clinical practice, and better understand the different approaches between specialists.

## Materials and methods

The survey was promoted on behalf of the Italian Radiation Oncologist Association (AIRO), as well as supported and endorsed by the Italian Pneumologists Association (AIPO) and Italian Thoracic Surgery Society (SICT). The questionnaire was divided into three sections: interviewer features, diagnostic management, and treatment approach.

An online, web-based survey was developed using a dedicated web platform (http://www.xsurvey.cloud). The survey was first planned in November 2016 and completed in April 2017. The questionnaire was written in Italian, and is provided in **S1 Appendix.**

Initially, all the written questions were submitted for analysis and approval by the Scientific Committee of AIRO (composed of national experts in clinical oncology). After initial AIRO Scientific Committee approval, the questions were then submitted to the AIRO Directory Board for final approval. Finally, in January of 2018, an initial e-mail with a link to the web-based questionnaire was sent to all AIRO members involved in thoracic oncology, and consequently to TS and PN dedicated to thoracic malignancy in daily clinical practice. The Medical Oncologists Society was also asked to forward the survey to physicians who manage lung cancer treatments. The survey was strictly confidential and anonymous, and designed to be completed in approximately 15 minutes. Initial responses were reviewed by AIRO Thoracic Oncology Group board members. A general reminder was sent one month later to obtain a greater rate of answers, and survey responses were collected until April 2018.

All responses (including partial responses) were deemed eligible and analyzed through descriptive statistics. Specifically, two statistical analyses were performed in order to investigate variability in answer distribution. First, subgroups were created from the total population based on specialization (PN, TS, radiation [RO] and medical oncologists [MO]), level of experience, workings hours spent on lung tumors issues, and rate of multidisciplinary case discussion. These subgroups were used to define current, clinical lung cancer practice in Italy by presenting interviewees with case studies for response. For this analysis, answers were given in a multi-choice format with no response considered "correct" or "incorrect".

Second, these subgroups were correlated with a "correct" response rate using the responses to the same set of case studies. "Correct" answers were defined as adhering to international guidelines (such as the ESMO) by the core of experts who planned the survey (**S2 Appendix**). Each subgroup was compared with the rest of the interviewed population in terms of answer distribution, indicating statistical significance when $p < 0.05$.

Both the above analyses were conducted using the Pearson chi-square test. As a survey involving neither therapeutic choice on humans nor demographic data, it did not require any ethics committee approval.

## Results

The survey was sent to 1,478 email-addresses. 421 responses were received (with an overall response rate of 28%), divided as follows: 176 (42%) RO, 86 (20%) MO, 92 (22%) PN, 64 (15%) TS and 3 (1%) other specialists. The features of all respondents are summarized in Table 1.

Regarding second level staging in LA-NSCLC, most respondents (63%) declared that a patient with a new LA-NSCLC diagnosis with mediastinal PET positivity required complete staging with endobronchial ultrasound-guided/trans-bronchial needle aspiration (EBUS/TBNA) (Fig 1). Interestingly, being a RO or TS seemed to positively influence the chance of reporting correct answers, as defined previously ($p = 0.001$ and $p = 0.0001$, respectively). Other characteristics with a statistically positive impact on the rate of correct answers were 10 to 15 years of experience ($p = 0.025$), the working hours spent on lung tumors ($\geq 50\%$), the presence of a weekly multidisciplinary team meeting ($p = 0.04$), and the number of treated patients (at least 10 per year, $p = 0.011$). EBUS/TBNA was the diagnostic standard choice for 57% of responders, even in case of lymph nodal mediastinal PET negativity. No other additional diagnostic tests were necessary for 99 colleagues (24%) (Fig 2).

Of note, almost half of the respondents (43%) considered at least a histological differential diagnosis between adenocarcinoma and squamous cell carcinoma mandatory before planning a radical treatment. On the other hand, 22% of lung specialists preferred to proceed with mutational analysis, and only 15% needed PDL-1 expression (question number 11). It is important to note that this survey was administered before the results of the PACIFIC trial [8].

**Table 1. Features of 421 respondents (AIRO/AIOM/AIPO/SICT) to the survey.**

| Characteristics | Response n (%) |
|---|---|
| *Type of specialization*: | |
| Radiation Oncology | 176 (42%) |
| Medical Oncology | 86 (20%) |
| Pneumology | 92 (22%) |
| Thoracic Surgery | 64 (15%) |
| Other | 3 (1%) |
| *Type of practice*: | |
| Academic | 165 (40%) |
| IRCCS | 53 (13%) |
| General hospital | 23 (6%) |
| Private | 23 (6%) |
| *Years in practice*: | |
| 0–5 years | 128 (31%) |
| 5–10 years | 51 (12%) |
| 10–15 years | 79 (19%) |
| > 15 years | 158 (38%) |
| *% of working hours spent on lung tumors* | |
| 90–100% | 53 (13%) |
| 70–90% | 85 (21%) |
| 50–70% | 108 (26%) |
| <50% | 168 (41%) |
| *Representative physician in diagnostic and staging process*: | |
| Radiation Oncologist | 7 (2%) |
| Medical Oncologist | 12 (27%) |
| Pneumologist | 208 (50%) |
| Thoracic Surgeon | |
| *Geographic location*: | |
| North | 182 (44%) |
| Central | 150 (36%) |
| South | 56 (14%) |
| Islands | 25 (6%) |
| *Rate of multidisciplinary case discussion*: | |
| Yes, with weekly meetings | 295 (72%) |
| Yes, with meetings every two weeks | 34 (8%) |
| Yes, but not regularly | 41 (10%) |
| None | 41 (10%) |
| *Number of patients with LA-NSCLC treated in the last year*: | |
| > 30 pts | 30 (34%) |
| 20–30 pts | 108 (26%) |
| 10–20 pts | 117 (29%) |
| <10 pts | 45 (11%) |

Dedicating less than 50% of the total working hours to lungs predisposed one to a higher rate of wrong answers to this question (p = 0.021).

Questions about therapeutic choices were presented in the form of clinical cases. Neoadjuvant chemotherapy (CT) followed by surgery was recommended by 43% of specialists (Fig 3)

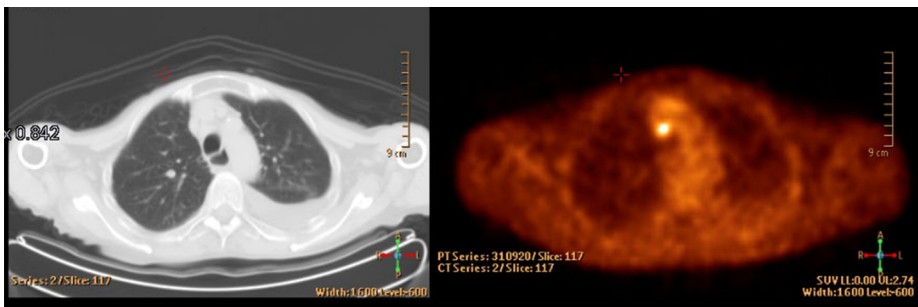

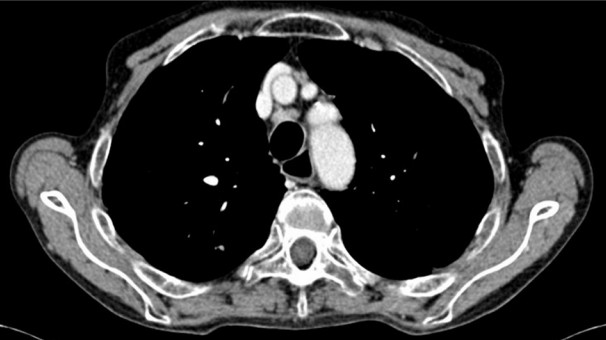

**Fig 1. LA-NSCLC lymph nodal mediastinal PET positivity–question 9.**

in a patient with cT1b cN2, single station involvement lung adenocarcinoma (stage IIIA), and fit for surgery. Instead, in a patient fit for surgery with clinical stage cT2 cN2 and multiple

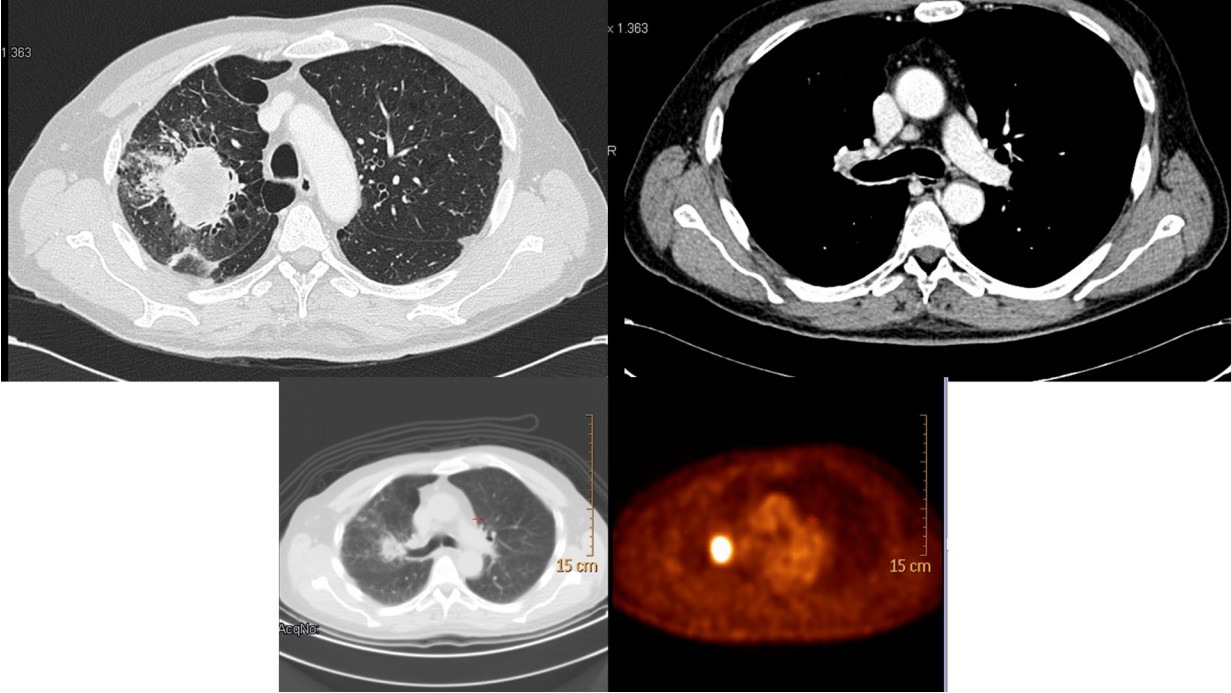

**Fig 2. LA-NSCLC diagnosis with lymph nodal negativity–question 10.**

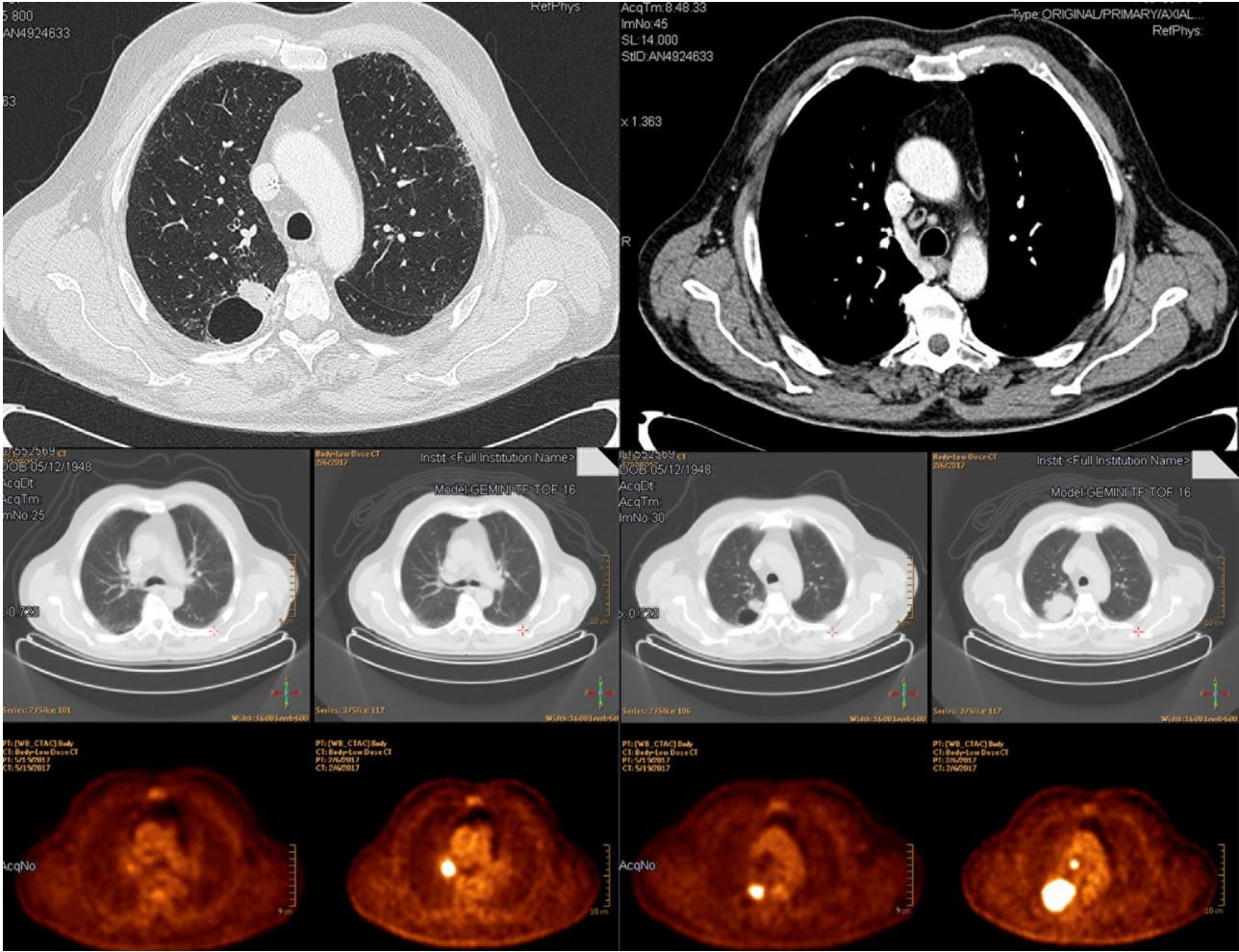

**Fig 3. LA-NSCLC clinical stage cT1b cN2, single station, fit for surgery—question number 12.**

positive lymph node stations (stage IIIA), 32% of respondents declared a preference for radical cCRT (Fig 4). Even in this case, experience with lung tumors was essential: more than 15 years of experience and dedicating more than 70% of the total working hours to lung cancer were positively correlated with a correct answer (p = 0.009 and p = 0.041, respectively). Almost half of respondents (48%) thought that radical cCRT, if not performed in a neoadjuvant setting, was the best approach for inoperable lung adenocarcinoma in partial response/stability (ycN2) after neoadjuvant chemotherapy. Surgery (only with a feasible lobectomy) and radiotherapy (RT) alone (if not administered in a neoadjuvant setting) were the best treatment options for 23% and 19% of lung specialists, respectively. The remaining part voted for CT alone or surgery, independently from the applied technique (Fig 5).

54% of respondents declared that upfront cCRT should be the preferred treatment for stage IIIA-B NSCLC patients; cCRT followed by consolidation CT was the best choice for 33%. Moreover, sequential CTRT was a reasonable treatment option for the remaining 13% of the interviewees (question number 15). For most respondents, acute toxicity and logistical barriers were reasonable contraindications for a concomitant approach in a patient fit for surgery with stage IIIA-B NSCLC or cCRT in stage IIIA-B (question number 16).

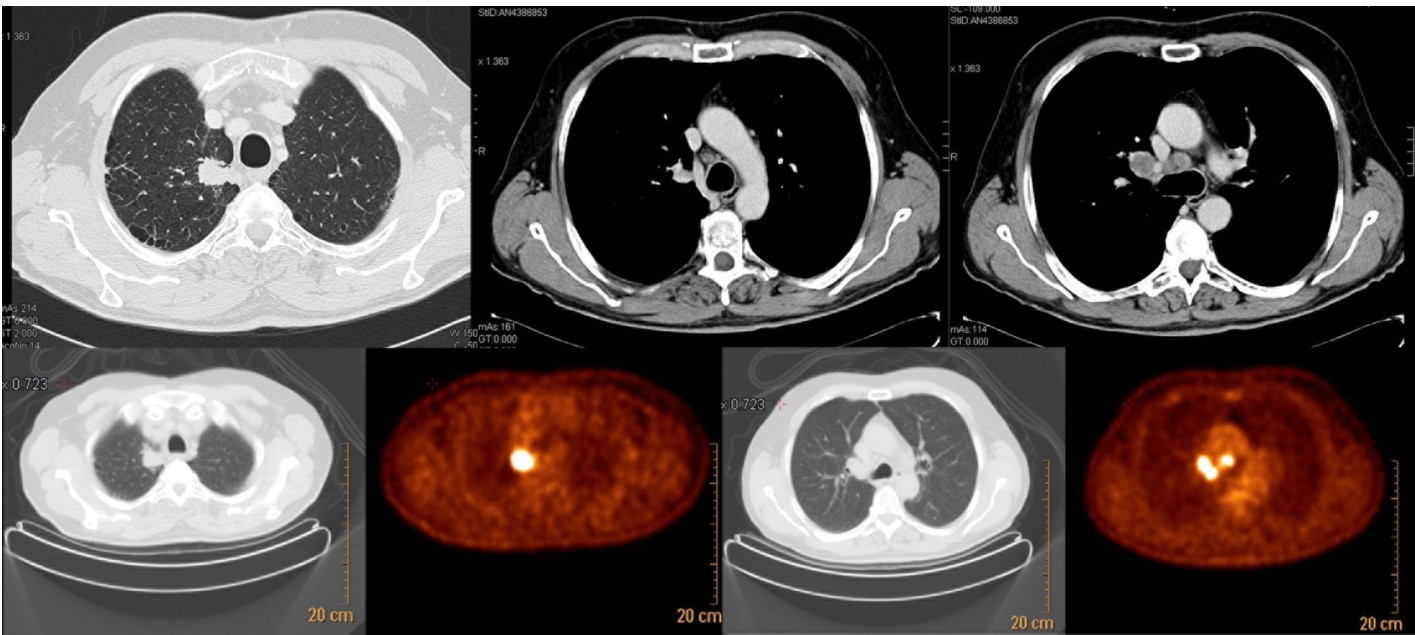

**Fig 4. LA-NSCLC clinical stage cT2cN2, multiple stations, IIIB—question number 13.**

After the first analysis, a significant difference was observed between the average response of the entire population and the various subgroups for each question analyzed both in diagnosis management and in therapeutic management (**S3 Appendix** and **S4 Appendix**).

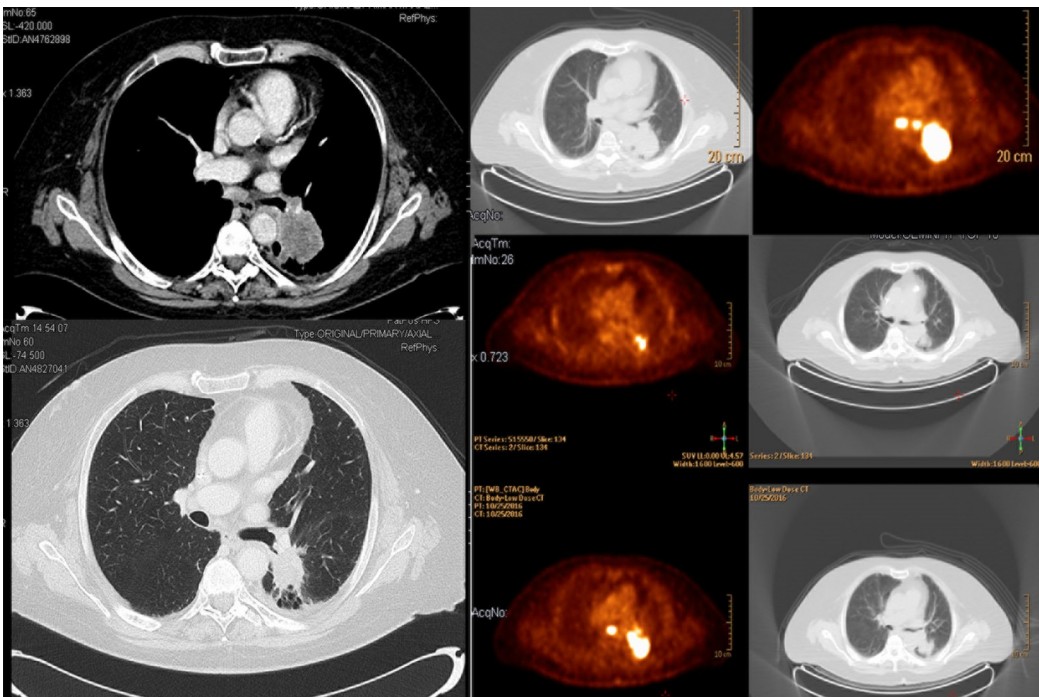

**Fig 5. LA-NSCLC inoperable in partial response/stability (ycN2) after neoadjuvant chemotherapy—question number 14.**

The results of the second statistical analysis comparing subgroups according to correct answers are shown in **S5 Appendix** and **S6 Appendix**. The p-value indicates whether a significant difference exists between each subgroup and the total population, and the percentage reflects the correct answer rate.

## Discussion

To the best of our knowledge, this is the first multidisciplinary survey in Europe that collected data about the management of Stage III NSCLC. Focusing on diagnostic approaches, we observed global agreement on the management of second level staging in mediastinal histo-pathological proof in accordance with ESMO (European Society of Medical Oncology) and ESTS (European Society of Thoracic Surgeons) clinical guidelines [9]. In particular, EBUS with FNA (fine-needle aspiration) was preferred over video-assisted mediastinoscopy (VAM) because it is minimally invasive and has acceptable sensitivity (83% and 94%) [10–11].

In our survey, over 70% of physicians agreed with an invasive histopathological proof in suspect mediastinal lymph nodes, and EBUS was considered the main invasive staging approach. The answers were conditioned by the availability of procedural equipment or resources.

Regarding histopathological and molecular analysis, nearly half of the physicians (43%) considered a diagnosis of adenocarcinoma or squamous cell carcinoma acceptable for a locoregional oncological approach. Surprisingly, 22% recommended a molecular analysis (EGFR, ROS, ALK1 status) in unresectable LA-NSCLC, in which platinum-based chemotherapy and radiotherapy represent the standard of care [12].

The PACIFIC trial demonstrated that durvalumab improved progression-free survival (HR 0.52; 95% CI, 0.42 to 0.65; P<0.001) [7] and overall survival (HR, 0.68; 99.73% CI, 0.47 to 0.997; p = 0.0025) [8] compared to placebo. This approach has since modified clinical management (excluding patients with a PD-L1 level < 1%), and the evaluation of PD-L1 status would be systematically requested in this setting. These findings could justify the answer given by 15% of survey responders, according to EMA approval.

Regarding diagnostic management, we observed some heterogeneity in the subgroup analysis, as reported in **S3** and **S4 Appendices**. In particular, there was a close correlation between high expertise, over >50% of working hours being dedicated to the management of LA-NSCLC, weekly multidisciplinary discussions, and good clinical practice.

The clinical guidelines recognize the importance of multidisciplinary meetings (MDM) in the management of lung cancer patients, as an early diagnosis can significantly impact survival, while the most appropriate treatments must be selected based on the clinical condition of the lung cancer patient [13, 14], especially in stage III NSCLC [15].

In general terms, dedicated lung cancer MDMs should feature the participation of key professionals: radiation oncologists, medical oncologists, thoracic surgeons, pneumologists, radiologists, nuclear medicine physicians, pathologists, and specialist nurses.

Another finding from this survey was the clinical management of different, locally advanced NSCLC disease, including potentially resectable and unresectable cases. In our survey, in cT1bN2 (Stage IIIA—single station), 43% of physicians proposed neoadjuvant CT followed by resection, while for 23%, upfront surgical resection was considered appropriate. These results underline that lobectomy and lymph node dissection are considered feasible in single N2 lymph node and limited T-stage cancer.

Certainly, surgery is accepted by the international community as the cornerstone approach in the management of lung cancer. In the ESMO guidelines [11], for patients with single N2 station involvement and early stage primary NSCLC, a lung resection followed by adjuvant

systemic therapy or neoadjuvant CT and surgery are both considered acceptable, even though it remains an open issue.

Regarding the correct approach for a patient with unresectable stage III NSCLC who had been treated with neoadjuvant CT and persistent N2 (ycN2), almost half of physicians (48%) considered cCRT the standard of care.

Finally, for cT2N2 (Stage IIIA—multiple stations, no bulky disease), there was diversity among the answers: 32% of physicians supported a radical cCRT treatment, 27% considered neoadjuvant CT as a feasible strategy, and 23% proposed surgery after CT or cCRT.

In both scenarios, cCRT (Level IA) is the most appropriate therapeutic approach according to ESMO guidelines. However, in centers with an experienced multidisciplinary team, a multi-modality approach including surgery can be considered in select cases (Level of Evidence IV C) [12].

Nevertheless, surgical resection could be considered even if different clinical experiences do not significantly impact clinical outcomes after a trimodality approach. In the Lung Intergroup 0139 trial, the induction of CT and RT followed by surgery did not prove advantageous for OS when compared to radical cCRT due to higher pneumonectomy-related toxicities. [16]. More recently, two additional studies explored the use of a neoadjuvant CTRT combination, but without a clear benefit for clinical outcomes [17, 18]. For these reasons, trimodality therapy cannot be considered the standard of care in locally advanced NSCLC.

When a locally advanced NSCLC patient was not considered eligible for surgery, upfront cCRT was considered the standard of care by 54% of physicians. This clinical approach agrees with the principal international guidelines and meta-analysis, confirming the advantage of cCRT compared to sequential RT [1, 12].

Despite the consideration of acute toxicity as the primary impetus for avoiding cCRT treatments, we still observed that in 8% of physicians, prescribing sequential C-RT was correlated to logistical limitations. Also, in the management of clinical cases, we observed a relationship between large clinical experience and good clinical practice, underlying the role of a multidisciplinary approach and expertise in LA-NSCLC management.

The results from the PACIFIC trial will modify daily clinical practice, and it is critical to improve education and multidisciplinary approaches in the management of LA-NSCLC to define the most appropriate diagnostic and therapeutic management strategy.

## Study limitations, strengths, and future perspectives

During the planning stage of the present survey, a discussion was held among the members of the AIRO Thoracic Oncology Group to clarify the main research goals. The promoters were aware of the reservations regarding surveys, which are generally a poor method for collecting data and opinions on current clinical practice. We could not verify the self-reported data, and respondents' memories are often unreliable. Moreover, we could not quantify practice outcomes, but we did obtain data on expert opinions, beliefs, and ideas regarding LA-NSCLC management in Italy. We now have a sense of the extent of agreement and disagreement among the various specialists regarding LA-NSCLC approaches. Such information could serve as the proof of principle for a consensus conference, designed to establish multidisciplinary indications for the staging and treatment of pulmonary LA-NSCLC. Moreover, this information could help to identify targets for future research projects and investigations.

## Supporting information

**S1 Appendix. Questionnaire.**
(DOCX)

**S2 Appendix. Correct answers accepted by NSCLC experts.**
(DOCX)

**S3 Appendix. Statistical analysis for diagnostic management comparing subgroups and entire population [table].**
(DOCX)

**S4 Appendix. Statistical analysis for therapeutic management comparing subgroups and entire population [table].**
(DOCX)

**S5 Appendix. Statistical analysis for diagnostic management comparing subgroups and correct answers [table].**
(DOCX)

**S6 Appendix. Statistical analysis for therapeutic management comparing subgroups and correct answers [table].**
(DOCX)

## Author Contributions

**Conceptualization:** Alessio Bruni, Niccolò Giaj-Levra, Carlo Greco, Davide Franceschini, Maria Taraborrelli, Silvia Novello, Vieri Scotti.

**Data curation:** Alessio Bruni, Niccolò Giaj-Levra, Patrizia Ciammella, Virginia Maragna, Katia Ferrari, Viola Bonti, Davide Franceschini, Marco Perna, Stefano Vagge, Editta Baldini, Andrea Botti, Silvia Novello, Vieri Scotti.

**Formal analysis:** Andrea Botti.

**Funding acquisition:** Vieri Scotti.

**Methodology:** Patrizia Ciammella, Andrea Botti.

**Supervision:** Alessio Bruni, Niccolò Giaj-Levra, Stefania Greco, Carlo Greco, Paolo Borghetti, Enrica Capelletto, Marco Perna, Giuseppe Banna, Stefano Vagge, Emilio Bria, Marcello Tiseo, Massimiliano Paci, Maria Taraborrelli, Venerino Poletti, Pierluigi Granone, Umberto Ricardi, Silvia Novello, Vieri Scotti.

**Validation:** Francesco Grossi, Giuseppe Banna, Silvia Novello, Vieri Scotti.

**Visualization:** Vieri Scotti.

**Writing – original draft:** Alessio Bruni, Niccolò Giaj-Levra, Patrizia Ciammella, Paolo Borghetti, Marco Perna, Andrea Botti, Silvia Novello, Vieri Scotti.

**Writing – review & editing:** Alessio Bruni, Niccolò Giaj-Levra, Patrizia Ciammella, Katia Ferrari, Paolo Borghetti, Davide Franceschini, Marco Perna, Silvia Novello, Vieri Scotti.

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
