## [Decision Letter · Decision Letter 0]

1 Aug 2019

PONE-D-19-18372

Management of Locally advanced Non Small Cell Lung Cancer in the modern era: a National Italian Survey on diagnosis, treatment and multidisciplinary approach

PLOS ONE

Dear Dr bruni,

Thank you for submitting your manuscript to PLOS ONE. After careful consideration, we feel that it has merit but does not fully meet PLOS ONE’s publication criteria as it currently stands. Specifically, the authors must improve the English grammar in the manuscript for it to be suitable for publication. Therefore, we invite you to submit a revised version of the manuscript that addresses the points raised during the review process.

We would appreciate receiving your revised manuscript by Sep 15 2019 11:59PM. To enhance the reproducibility of your results, we recommend that if applicable you deposit your laboratory protocols in protocols.io, where a protocol can be assigned its own identifier (DOI) such that it can be cited independently in the future. For instructions see: http://journals.plos.org/plosone/s/submission-guidelines#loc-laboratory-protocols

We look forward to receiving your revised manuscript.

Kind regards,

Stephen Chun

Academic Editor

PLOS ONE

Journal Requirements:

2. Please remove the tracked changes from page 6 of your manuscript.

 [The funders had no role in study design, data collection and analysis, decision to publish, or preparation of the manuscript.]

Please provide an amended Funding Statement that declares *all* the funding or sources of support received during this specific study (whether external or internal to your organization) as detailed online in our guide for authors at http://journals.plos.org/plosone/s/submit-now.  

Please state what role the funders took in the study.  If any authors received a salary from any of your funders, please state which authors and which funder. If the funders had no role, please state: "The funders had no role in study design, data collection and analysis, decision to publish, or preparation of the manuscript."

*Please include your amended statements within your cover letter; we will change the online submission form on your behalf.

4. Please include a copy of Tables 2 and 3 which you refer to in your text on page 13.

Reviewers' comments:

Reviewer's Responses to Questions

**Comments to the Author**

1. Is the manuscript technically sound, and do the data support the conclusions?

Reviewer #1: Yes

Reviewer #2: Partly

Reviewer #3: Yes

2. Has the statistical analysis been performed appropriately and rigorously? 

Reviewer #1: Yes

Reviewer #2: Yes

Reviewer #3: Yes

3. Have the authors made all data underlying the findings in their manuscript fully available?

Reviewer #1: Yes

Reviewer #2: Yes

Reviewer #3: Yes

4. Is the manuscript presented in an intelligible fashion and written in standard English?

Reviewer #1: No

Reviewer #2: No

Reviewer #3: No

5. Review Comments to the Author

Reviewer #1: This is a survey of several multidisciplinary members of a thoracic oncology team regarding management of LA-NSCLC. Comments to the authors follow.

- Complete English language review recommended by a native speaker. There are too many grammatical and tense errors in the document to correct.

- Line 105: "9617" should be "0617"

- If you are making a discussion of what "fit" means, perhaps it's best to state the types of patients enrolled on those trials like 0617 and PACIFIC. (e.g. performance status, age, etc)

- Please state the overall response rate of the survey

- Lines 168-170: confusing, please reword.

- Please comment in the Discussion why so many responders may have wanted a diagnosis of adeno vs SCC beforehand, since that usually does not change management. What about the diagnosis "NSCLC not otherwise specified" in which the pathologist cannot tell if it's adeno or SCC?

- The Discussion is long and the authors should make efforts to streamline this information better.

Reviewer #2: This is an interesting survey report from the University Hospital of Modena in Italy.

It seeks to gather information from different specialists (radiation oncologists, medical oncologists, pulmonologists, and thoracic surgeons) who treat locally advanced lung cancer in Italy. The survey uses 15 questions regarding both the best way to diagnose/stage and treat this cancer. The cases presented use imaging which is quite good. The survey questions were reviewed by the scientific committee of the Italian Radiation Oncologist Association. The authors evaluated responses as either “correct” or “not correct” based on “main international guidelines” (ESMO) and “following the rules of good clinical practice” by the core of experts who designed the survey. Overall, I liked this concept though “following the rules of good clinical practice” probably shouldn’t be suggested as acceptable for peer review.

I believe there is valuable data here.

My main problem with the manuscript has to do with its presentation

For example, what’s going on with page 6 lines 123-128?

It’s good to edit your manuscript, but do this before submitting it for peer review.

Finally, the American grammar is just not acceptable for publication in an American journal.

A few examples:

1) Page 12 line 229 “is preferred because minimally invasive” should be “is preferred because it is minimally invasive”.

2) Page 13 line 247 “will be systematically request in this setting” should be “will be systematically requested in this setting”.

3) Page 13 line 250 “disomogeneity” is not a word.

4) Page 15 line 298 “this negative result should be related to a higher toxicity” should be “this negative result is related to a higher toxicity”.

Reviewer #3: The authors are to be commended for seeking multidisciplinary perspectives on the management of Stage III NSCLC. The paper describes survey results. It would be helpful to know how correct answers were chosen. The paper also contains many grammatical and vocabulary errors and the paper would benefit from additional writing assistance.

6. PLOS authors have the option to publish the peer review history of their article (what does this mean?). If published, this will include your full peer review and any attached files.

Reviewer #1: No

Reviewer #2: Yes: John M. Holland, MD

Reviewer #3: No

---

## [Author Response · Author response to Decision Letter 0]

27 Aug 2019

Dear Editor,

thank you for the opportunity to revise and resubmit our manuscript entitled “Management of locally advanced non small cell lung cancer in the modern era: a national Italian survey on diagnosis, treatment and multidisciplinary approach”. According the reviewers advices, we revised our manuscript and highlighted all changes in the manuscript. The response to the reviewers are listed as follows. We hope that our modifications would meet your requirements. 

We look forward to hearing from you, and are hopeful of a positive response. 

Should you have any questions, please feel free to contact me.

Thank you and best regards.

Editor

I removed the tracked changes from page 6 of your manuscript

At page 13, I replaced “Table 2” and “Table 3” with “S3 Appendix” and “S4 Appendix” due to the complexity of the tables reported (as you suggested in a previous review).

I included captions for the supporting information files at the end of the manuscript, and I updated any in-text citations to match accordingly.

Reviewer #1

I submitted the paper to a native speaker to have a complete English language review and to correct all the grammatical and tense errors 

To better explain what “fit” means, I maintained what we proposed as definition and I also reported the main features of patients enrolled in RTOG0617.

I added the overall response rate (in percentage of responders) of the survey at the beginning of graph “Results”

I reworded the Lines 168-170 that were confusing.

Together with all the co-aouthors, I tried to reduce the lenght of the graph “Discussion” underlying the most significative findings.

Reviewer #2

I modified the tense at page 6 line 123-128 trying to make it clearer

I accepted all the corrections/suggestions :

Page 12 line 229 

Page 13 line 247 

Page 13 line 250 

Page 15 line 298 

Reviewer #3

We reported in the text that answers were assumed as “correct” by the experts taking into account the main International Guidelines such as ESMO ones.

Waiting for hearing from you,

kind regards

Alessio Bruni, MD

---

## [Decision Letter · Decision Letter 1]

30 Aug 2019

PONE-D-19-18372R1

Management of Locally advanced Non Small Cell Lung Cancer in the modern era: a National Italian Survey on diagnosis, treatment and multidisciplinary approach

PLOS ONE

Dear Dr Bruni,

Thank you for submitting your manuscript to PLOS ONE. After careful consideration, we feel that it has merit but does not fully meet PLOS ONE’s publication criteria as it currently stands. Therefore, we invite you to submit a revised version of the manuscript that addresses the points raised during the review process.

We would appreciate receiving your revised manuscript by Oct 14 2019 11:59PM. To enhance the reproducibility of your results, we recommend that if applicable you deposit your laboratory protocols in protocols.io, where a protocol can be assigned its own identifier (DOI) such that it can be cited independently in the future. For instructions see: http://journals.plos.org/plosone/s/submission-guidelines#loc-laboratory-protocols

We look forward to receiving your revised manuscript.

Kind regards,

Stephen Chun

Academic Editor

PLOS ONE

Additional Editor Comments (if provided):

The manuscript is improved, but continues to have linguistic issues that must be addressed prior to publication.

Reviewers' comments:

Reviewer's Responses to Questions

**Comments to the Author**

1. If the authors have adequately addressed your comments raised in a previous round of review and you feel that this manuscript is now acceptable for publication, you may indicate that here to bypass the “Comments to the Author” section, enter your conflict of interest statement in the “Confidential to Editor” section, and submit your "Accept" recommendation.

Reviewer #1: All comments have been addressed

Reviewer #2: All comments have been addressed

2. Is the manuscript technically sound, and do the data support the conclusions?

Reviewer #1: Yes

Reviewer #2: Yes

3. Has the statistical analysis been performed appropriately and rigorously? 

Reviewer #1: Yes

Reviewer #2: Yes

4. Have the authors made all data underlying the findings in their manuscript fully available?

Reviewer #1: Yes

Reviewer #2: Yes

5. Is the manuscript presented in an intelligible fashion and written in standard English?

Reviewer #1: Yes

Reviewer #2: (No Response)

6. Review Comments to the Author

Reviewer #1: The authors have made substantial strides to revise their paper, but there are still some English language issues. E.g. "Both the above analyses were conducted using Person chi-square test. As a survey not involving either therapeutic choice on humans or demographic data, it does not need any ethics committee approval."

- "the Pearson" not "Person"

- 2nd sentence should be past tense like the rest

E.g. "thought a radical cCRT" remove 'a'

Please have the native English speaker carefully look over the English, tense, usage, and flow one more time carefully

Reviewer #2: (No Response)

7. PLOS authors have the option to publish the peer review history of their article (what does this mean?). If published, this will include your full peer review and any attached files.

Reviewer #1: No

Reviewer #2: No

---

## [Author Response · Author response to Decision Letter 1]

3 Oct 2019

Dear Editor,

thank you again for the opportunity to revise and resubmit our manuscript entitled “Management of locally advanced non small cell lung cancer in the modern era: a national Italian survey on diagnosis, treatment and multidisciplinary approach”. According the reviewers advices, we revised again our manuscript and highlighted all changes in the manuscript. 

The response to the reviewers are listed as follows. 

We hope that our modifications would meet your requirements. 

I re-submitted the paper to a native speaker to have a complete English language review and to correct all the grammatical and tense errors 

We accepted all the corrections/suggestions:

- “Person" was changed in “the Pearson"

 - “ […]it does not need any ethics committee approval." Was changed in “[..] it did not need any ethics committee approval”

 - "thought a radical cCRT" was changed in "thought radical cCRT"

 - Other imprecisions were again corrected by our native speaker reviewer

We look forward to hearing from you, and are hopeful of a positive response. 

Should you have any questions, please feel free to contact me.

Thank you and best regards.

Alessio Bruni, MD

---

## [Editor Report · Decision Letter 2]

4 Oct 2019

Management of Locally advanced Non Small Cell Lung Cancer in the modern era: a National Italian Survey on diagnosis, treatment and multidisciplinary approach

PONE-D-19-18372R2

Dear Dr. Bruni,

We are pleased to inform you that your manuscript has been judged scientifically suitable for publication and will be formally accepted for publication once it complies with all outstanding technical requirements.

With kind regards,

Stephen Chun

Academic Editor

PLOS ONE

Additional Editor Comments (optional):

While the manuscript is accepted, there remain minor issues with grammar and flow. In the publication process, the authors are expected to further improve the English of the manuscript.
---

## [Editor Report · Acceptance letter]

21 Oct 2019

PONE-D-19-18372R2 

Management of locally advanced non-small cell lung cancer in the modern era: a national Italian survey on diagnosis, treatment and multidisciplinary approach. 

Dear Dr. Bruni:

I am pleased to inform you that your manuscript has been deemed suitable for publication in PLOS ONE. Congratulations! Your manuscript is now with our production department. 

With kind regards,

on behalf of

Dr. Stephen Chun 

Academic Editor

PLOS ONE